# Factors Associated with Coping Behaviors of Abused Women: Findings from the 2016 Domestic Violence Survey

**DOI:** 10.3390/healthcare10040622

**Published:** 2022-03-25

**Authors:** Youngran Han, Heejung Kim, Nawon An

**Affiliations:** 1College of Nursing, Gyeongju Campus, Dongguk University, Gyeongju 38066, Korea; hanyr@dongguk.ac.kr; 2Mo-Im Kim Nursing Research Institute, College of Nursing, Yonsei University, Seoul 30722, Korea; hkim80@yuhs.ac; 3Department of Nursing, Cheongam College, Suncheon 57997, Korea

**Keywords:** intimate partner violence, adaptation, psychological, public health

## Abstract

Background: Intimate partner violence (IPV) is an important public health problem. In Korea, limited studies have systematically investigated the coping strategies used by female IPV victims. Purpose: We identified the factors associated with abused women’s coping behaviors in South Korea. Methods: This descriptive, cross-sectional study comprises secondary data analysis using the 2016 Domestic Violence Survey; we examined women who experienced domestic violence (DV) in the last year: September to December 2016 (*n* = 309). Multinomial logistic regression analyses were conducted using an ecological model. Results: Over 50% responded they “did nothing”, while others “escaped the scene of violence or ran outside” and “became reciprocally violent” as coping behaviors sequentially. Compared to the women who “did nothing”, women who experienced feelings of intimidation or fear due to DV, were sexually abused, and suffered physical injury were 5.44, 3.22, and 3.02 times, respectively, more likely to escape from the scene than those who did not. Most showed passive coping behaviors. Relationship level factors, such as type of DV and physical injury, were significantly associated with the type of coping behavior. Conclusions: Our findings emphasize that multi-level comprehensive health programs are required, especially for women coping passively, to prevent and respond to DV.

## 1. Introduction

Intimate partner violence (IPV) is a public health priority that has a negative impact on the victim, their children, and family members. In addition, it could lead to physical and psychological health problems among the victims, resulting in the intergenerational transmission of violence [1]. Although both men and women can be IPV victims and perpetrators, IPV against women refers to any violent behavior perpetrated by a current or former male intimate partner that causes physical, sexual, or psychological harm within the context of marriage, cohabitation, or any other formal or informal union [2]. Violence perpetrated by husbands against their wives is the most common form of IPV. It poses a significant problem that has both short- and long-term physical, psychological, and social impacts due to its persistence [3]. An analysis of data on the prevalence of IPV between 2000 and 2018 from 161 countries and areas revealed that 26% of girls and women aged 15 years and older had been subjected to physical and/or sexual violence by a current or former husband or male intimate partner within the past 12 months [2]. The distribution of IPV shows large differences, with its prevalence being up to 4% in high-income countries and at least 25% in some low-income countries [2]. Further research is required on the factors that explain the differences in the prevalence of partner violence across countries.

The South Korean society has long tolerated domestic violence (DV) due to its patriarchal, male-dominated, and family-centric culture [4]. Gender roles have changed recently due to rapid modernization and westernization in South Korea, altering the perception of DV [4]. According to a survey conducted every three years by a Korean government agency, the Ministry of Gender Equality and Family (MGEF), the prevalence of female DV victims in South Korea has been consistently high (33.1% in 2007, 39.1% in 2010, and 29.8% in 2013) [3,5].

IPV remains hidden, stigmatized, and largely unrecognized by healthcare service providers [6]. However, abused women often require health services, typically making healthcare providers their first point of professional contact [6]. The health sector plays an important role in serving as a point of contact for early identification of IPV and providing inclusive healthcare to these women. Furthermore, these women can then be directed to other appropriate support services [6].

Coping strategies are appraisals and behaviors that use internal or external resources to control distress when one’s stress exceeds tolerable levels [7]. In general, they are classified as active versus passive or problem-focused versus emotion-focused [8]. The coping strategies used by victims of repeated DV can deeply affect their psychological function and physical well-being. Thus, they are considered essential [9,10]. Many studies on the coping strategies of female IPV victims, such as surveys on the coping strategies or perceptions of DV victims in rural areas, are conducted by agencies [11,12,13]. Additionally, there are literature-analysis-based studies on the predictive factors and outcomes of coping strategies [14], coping related to outcomes such as PTSD [10,15], and service interventions to improve coping strategies [9]. Although coping is an important topic in IPV research, limited studies in South Korea have systematically investigated the coping strategies used by female victims of DV [16]. It is impossible to customize the content of the services provided to meet the victims’ needs without first understanding their coping strategies. Therefore, further research focusing on the types of coping strategies used by female victims of DV to cope with violence is essential [12].

Our research question was what coping strategies do female DV victims in South Korea use and what factors influence those coping strategies? The ecological model conceptualizes violence as a multidimensional phenomenon based on the interaction of personal, situational, and sociocultural factors. It may also be applicable to DV [1,6]. Hobfoll (1986) stated that ecological perspectives aided the understanding of the interconnection between personal, environmental, and situational factors in the coping strategies of IPV victims [17]. Based on previous research [1,6], our study included the individual and their relationship, community, and societal level in the ecological model. Detailed information is presented in Figure 1.

IPV against women has been imperceptible in national and international statistics. Therefore, it is necessary to conduct a baseline for its prevalence through a population-based survey and conduct an analysis [6]. The Domestic Violence Survey (DVS), a triennial study, has been conducted by MGEF since 2007, according to the Domestic Violence Prevention and Victims Protection Act [18]. Thus, this study conducted secondary data analysis (SDA) using data from the 2016 National Survey based on an ecological model. The subjects in this study were females who had recently experienced DV as per the 2016 DVS. Among the variables investigated in the 2016 DVS, variables included in each level of the ecological model were selected. This study aims to identify the factors associated with the coping behavior of women who had experienced DV. The specific aims are (a) to examine the rate of DV by type and understand the damage or harm inflicted on the victims, as well as their coping behavior, and (b) identify the factors associated with the types of coping behaviors. The findings of this study may provide basic data for evidence-based, comprehensive programs that can help respond to and prevent DV.

## 2. Materials and Methods

### 2.1. Design, Data, and Study Samples

This was a descriptive, cross-sectional study. The 2016 DVS data, the primary data of this study, is the only large-scale national survey of DV in South Korea. From 22 September 2016 to 8 December 2016, 600 survey districts were selected using probability proportion sampling to represent the whole country, and 10 households were sampled systematically from each sampled survey district. Subsequently, one member (over 19 years old) in each household was selected as the subject of the survey. Finally, the survey was conducted on 6000 adults (4000 females, 2000 males) 19 years of age or older. DVS data were released to researchers in early 2019 [19]. The women in this SDA were those who had recently experienced DV (*n* = 309). Eligibility criteria included (1) being female and (2) having experienced DV during the previous year. The exclusion criteria were women who did not respond to the questions (*n* = 1), provided incorrect answers (*n* = 2), or asked others for help (*n* = 3, see Figure 2).

### 2.2. Variables

#### 2.2.1. Dependent Variable

The participants’ coping behavior, the dependent variable, was measured as a response at the time of DV. The responses included “did nothing”, “escaped the scene of violence or ran outside”, “became reciprocally violent”, and “asked others for help”. Of these, the response “asked others for help” was excluded from the analysis due to the small number of respondents (*n* = 3).

#### 2.2.2. Independent Variables

The independent variables consisted of four levels according to the ecological model (Figure 1).

##### Individual Level

For “childhood maltreatment by parents”, the response was classified as “Yes” if the subject had experienced one or more of six items related to physical and psychological abuse as well as child negligence by parents when they were 18 years or younger. For “witnessing violence between parents in childhood”, the response was also classified as “Yes” if the subject had witnessed one or more of three items about verbal or physical violence between their parents.

##### Relationship Level

At the relationship level, the structure of decision-making between the partners was enquired to establish who made the decisions in four areas (living expenses, child-rearing and education, house purchasing and moving, and investment and property management). Responses were rated on a 5-point Likert scale: each score was assigned from −2 (if the partner was responsible for all the decisions) to +2 (if the subject was responsible for all decisions), and if decisions were decided together through a discussion, the score would be close to 0. Cronbach’s α of this scale was 0.72 in the 2016 DVS; in this study, it was 0.75. Domestic violence was categorized into four types: psychological, physical, economic, and sexual, after revising and complementing the expressions and terms of 15 behaviors by reviewing the original version of the Revised Conflict Tactics Scale (CTS2) [20]. The CTS2 comprised seven items on physical violence, three items on psychological violence, three items on economic violence, and two items on sexual violence. Cronbach’s α of this scale was 0.74 in the 2016 DVS; in this study, it was 0.63. In this study, if the subject suffered at least one or more items of each type of violence over the past year (August 2015 to July 2016), the experience of violence by type was regarded as “Yes”. Physical injury, psychological suffering, and feelings of fear and intimidation experienced due to DV in their lifetime were measured on a 4-point Likert scale on the original instrument. However, in this study, the responses were classified as “No” for no wound/did not feel intimidation or fear and “Yes” for responses which ranged from suffering minor to severe injuries and from feelings of slight to severe intimidation or fear.

##### Community Level

Awareness of the neighborhood and community at the community level was measured via eight items related to relationships with neighbors and perception of public safety on a 4-point Likert scale. These comprised the following components: “There are many people in my neighborhood whom I can depend on emotionally”, “I can ask for help or consult my neighbors in case of a problem”, “I often help or receive help from my neighbors”, “If there is a fight, my neighbors will likely step up and intervene”, and “The local police are doing well on patrol”. Possible scores on the scale range from “not at all” (1 point) to “very true” (4 points). Higher average scores indicated a more positive perception of the community. Cronbach’s α of this scale was 0.86 in the 2016 DVS; in this study, it was 0.84.

##### Societal Level

The societal level consisted of four variables. The attitude toward gender roles was measured via seven items on a 4-point Likert scale. Higher average scores represented more patriarchal attitudes toward gender roles. The items included whether the role of the man was to be a leader in society and lead sexual relationships, whether wives should mostly do housework and obey their husband’s decision about whether they should have a job, and so on. Cronbach’s α of this scale was 0.87 in the 2016 DVS; in this study, it was 0.83. Attitude toward DV was measured via 10 items on a 4-point Likert scale. Higher scores indicated higher tolerance for DV. The items were composed of whether one can be violent to one’s family if one loses control in anger, is experiencing excessive stress, or is intoxicated; whether an abuser can be forgiven if they genuinely repent after the violence; whether intimate violence is a within-family problem; whether women should endure the violence to protect their family, and so on. Cronbach’s α of this scale was 0.71 in the 2016 DVS; in this study, it was 0.62. Awareness of DV-related laws and policies and supporting facilities consisted of six items, and the scores were assigned based on the subject’s recognition of each item: 1-point for recognition and 0-point for non-recognition. The closer the average score was to 1-point, the higher the level of recognition. The items on the awareness of IPV-related laws and policies include “Non-family members can report to police if domestic violence is suspected”, “There is guaranteed confidentiality for the person reporting”, and “Police must respond immediately on-scene to investigate upon receiving a report of domestic violence”, among others. Cronbach’s α of this scale was 0.77 in the 2016 DVS; in this study, it was 0.72. The six components regarding awareness of supporting facilities included the DV Counselling Center, shelter for DV victims, the 1366 Hotline (Korea Women’s Hotline), and so on. Cronbach’s α of this scale was 0.77 in the 2016 DVS; in this study, it was 0.70.

#### 2.2.3. Validity and Reliability

The 2016 DVS is a national statistical survey by the MGEF. To enhance the reliability of the survey as official statistics, a comprehensive quality assessment on the entire statistical process, from production to dissemination of official statistics, was conducted [19].

### 2.3. Ethical Considerations

The DVS data were collected based on the World Health Organization guidelines of ethics and safety related to the research and survey of violence against women [21]. This study was conducted in accordance with the Declaration of Helsinki. The Institutional Review Board (IRB) of the affiliated university approved the exempt status of the SDA (IRB approval No.: DGU IRB 20190020-01).

### 2.4. Data Analysis

All data analyses were conducted using IBM SPSS (version 23.0; SPSS Corp., College Station, TX, USA). The weighted data used in the analysis were calculated through three processes: design weight (strata and cluster), adjustment to non-response, and adjustment of population information [21]. Descriptive statistics based on the frequencies and weighted percentages were obtained to describe the socio-demographic variables and other DV-related data. Furthermore, a one-way analysis of variance (ANOVA) and chi-squared tests based on the means and weighted percentages were conducted to estimate the differences in coping behaviors according to each level of the ecological model. Subsequently, multinomial logistic regression analyses were conducted to identify the association between the type of coping behaviors for DV and the selected factors based on the ecological model. The results were presented as *p*-values and 95% confidence intervals (CIs).

## 3. Results

Of the 309 women who reported being abused by their husbands in the past year, the most frequent age group was 40–59 years (57.5%) (Table 1). According to the type of DV, 81.2% experienced psychological violence, 46.4% experienced physical violence, 20.0% experienced economic violence, and 18.4% experienced sexual violence. Regarding coping strategies, 64.8% (198 individuals) “did nothing”, 25.1% (77 individuals) “escaped the scene of violence or ran outside”, and 10.1% (34 individuals) “became reciprocally violent” (Table 1, Figure 3).

Table 1 shows the differences in the multi-level factors based on the three coping behaviors. There were significant group differences regarding age (*p* = 0.047), physical violence (*p* < 0.001), and sexual violence (*p* = 0.019) in the previous year. Although the proportion of victims who “did nothing” was highest across all age groups and types of violence, those aged 40–59 or over the age of 60 years were more likely to escape the scene of violence or run outside rather than become reciprocally violent. However, when compared with the group over 45 years, the 19–39 age group was more likely to become reciprocally violent than to avoid the scene.

There were significant group differences regarding physical violence (*p* < 0.001), sexual violence (*p* = 0.019), physical injury (*p* < 0.001), psychological distress (*p* < 0.001), and feelings of intimidation or fear due to DV in the previous year (*p* < 0.001). Among those who experienced psychological distress, the proportion of those who “did nothing” was highest at 47.0%, followed by 41.2% for those who avoided the scene. However, 56.2% of those who suffered physical injuries due to DV escaped the scene, which comprised the largest proportion of coping strategies. High proportions (over 40%) of those who experienced intimidation or fear during the DV episode did nothing or escaped the scene of violence (Table 1, Figure 3).

As the highest proportion of coping strategies observed was “did nothing”, an additional analysis was conducted to examine the reasons why participants may have reacted this way. They reported several reasons regarding why they did not respond to episodes of DV: (1) they thought they only needed to endure it for the moment, (2) they stated that “the perpetrator was my partner”, (3) they felt ashamed, and (4) they felt afraid that the violence would worsen if they responded.

Furthermore, an additional analysis was conducted on the experience of participants who asked for help, which was excluded from the analysis of coping strategies, as only three participants reported asking for help during a DV incident. Among the 309 abused women, only 20.4% asked for external help at the time of and after experiencing DV. They mostly reached out to their family, relatives, neighbors, or friends for help (see Table 2).

Table 3 shows the results of the multinomial logistic regression analysis. Several factors associated with the three different coping behaviors were identified with pseudo R^2^ = 0.206–0.364 (*p* < 0.001). When the “did nothing” group was set as a reference group to identify coping behaviors, it was found that women who were sexually abused by their partners were 3.22 times more likely to run away from the scene than were those who were not (OR = 3.22, 95% CI: 1.24–8.37, *p* = 0.016). Similarly, women who suffered physical injury from DV and experienced feelings of intimidation or fear due to DV were 3.08 (OR = 3.08, 95% CI: 1.26–7.52, *p* = 0.014) and 5.44 times (OR = 5.44, 95% CI: 2.16–13.70, *p* < 0.001), respectively, more likely to run away from the scene than were those who had not. However, there were no significant risk factors related to reciprocal violence (Table 3, Figure 3).

## 4. Discussion

This study has identified the prevalence of the different types of domestic violence among 309 abused women based on the 2016 DVS data. Furthermore, we attempted to comprehend the factors associated with their coping behaviors. We examined the types of DV and found that the rate of emotional violence was the highest, followed by physical violence and economic and sexual violence. Due to female victims of DV being likely to suffer more than one type of abuse, duplicate responses were possible. Based on reports from the 2007, 2010, 2013, and 2016 DVS data, among all female participants, 10.5–33.6% were emotionally abused, 3.3–15.3% were physically abused, 2.4–3.5% were financially abused, and 2.3–9.5% were sexually abused [3,5]. A direct comparison of such data with the findings of our study, which involved only women who experienced DV during the previous year of the DVS, was difficult. Nevertheless, the proportion of emotional abuse was highest across all four time points, followed by physical abuse. Financial and sexual abuse constituted the smallest proportion, which demonstrated the same trend by the type of DV.

In the global prevalence estimates of IPV, up to 4% of females aged 15 years and older in many high-income countries have experienced physical and/or sexual violence perpetrated by a husband or male intimate partner within the past 12 months [2]. These results were similar to those of the 2016 DVS results. Among the total female respondents, 3.3% and 2.3% of women experienced physical and sexual violence, respectively, from their husband or a man they lived with within the past 12 months.

Regarding DV-related coping behaviors in the 2010, 2013, and 2016 DVSs, the most frequent coping behavior was consistently “did nothing” (33.7%, 66.4%, and 64.8%), followed by “escaped the scene of violence or ran outside” (29%, 17.5%, and 25.1%), and “became reciprocally violent” (27.5%, 13.1%, and 10.1%), respectively [3]. According to the 2016 DVS, the reasons why victims chose to “do nothing” were because: the perpetrator was their husband, they thought of their children, they thought they needed to endure the DV only at the moment, and feeling embarrassed [21]. These reasons were likely a manifestation of the South Korean family-oriented culture of trying to keep the family together without revealing the occurrence of DV to others [22,23]. This can also be interpreted as an attempt to avoid the current situation [15], which can be understood as a passive or avoidant coping method. A study of abused women in Hong Kong, a culture similar to South Korea, where Confucianism is prevalent, also reported that women were likely to remain silent and endure the pain if the violence was not severe [24]. Meanwhile, “the victim’s perception of their relationships” was found to impact coping behaviors [25,26]. That is, if the female victims wish to maintain a relationship with the perpetrator, they are likely to endure the situation, which reflects the characteristics of DV [26]. These results support the findings of our study.

The “escaped the scene of violence or ran outside” and “became reciprocally violent” coping behaviors were reported by approximately one-third of the victims. A study on abused rural women reported that resistance strategies, such as running away and fighting, helped them secure safety [12]. Leaving the scene of violence was the most general coping behavior to protect oneself both when and before the violence occurs [12,13]. While some consider this a passive and ineffective coping strategy, female victims tend to consider this very effective and appropriate. Fighting men can be disadvantageous, especially because women are often physically weaker than men. Furthermore, fighting a husband can be viewed as immoral for a wife due to cultural factors [13]. However, it is necessary to secure the safety of women upon their return home after a short leave period, owing to financial and social poverty and the possibility of greater violence in the future [27]. Shelters for DV victims provide accommodation, meals, and counseling-oriented services [22], and private supporters such as friends and family provide emotional support depending on their level of DV awareness [28]. There are few financial aids available, and countermeasures are also required [22]. In this study, some abused women responded that they “became reciprocally violent” as a coping strategy. This behavior can be regarded as a resistive behavioral coping strategy [12]. However, this strategy requires careful consideration since compared to non-reciprocal IPV, it increases the risk of violence and injury [27,29]. Abused women are called victims of chronic victimization as they persistently experience DV in their homes. They experience fear repeatedly in abusive relationships and feel helpless in response. Therefore, abused women tend to exhibit emotional and reactive coping strategies rather than problem-solving approaches [30]. Iverson et al. (2013) stated that passive strategies may be helpful in reducing the overwhelming and painful emotions of individuals in the short term. However, they can increase the vulnerability of abused women by making it difficult for them to access their internal and external resources [10]. Passive strategies are known to cause dissociation, post-traumatic stress disorder, and revictimization. Therefore, problem-solving coping strategies and seeking social support are considered more effective coping strategies [10].

Next, compared to the women group who “did nothing”, women who experienced feelings of intimidation or fear due to DV, were sexually abused, or suffered physical injury from DV were more likely to run away from the scene. A study on abused women found that they exhibited an “avoid the moment only or run away from home” coping behavior when they required medical treatment due to physical violence [22], which was consistent with the findings of our study. Many studies showed that abused women acted passively and privately until they were no longer able to tolerate DV and went to an emergency room or shelter [15,24,31]. These results support the findings of our study. Meanwhile, recognition of serious danger was identified as a factor that influenced the support selection of women who were victims of DV. This partially supported the findings of this study, where a greater perceived threat in a close relationship leads to a more active search for external help [32]. Coping behaviors for sexual violence show inconsistent results among different studies, indicating a need for further research to better understand the choice of coping behavior depending on the type of violence [12,15]. In conclusion, when the victims could not tolerate the DV, they “escaped the scene of violence or ran outside”. This was a temporary strategy for female victims’ safety, not an active coping behavior to solve the problem. However, it was different from “did nothing”. These strategies do not alleviate the root cause of IPV. The avoidant response was found to encourage partner violence and exacerbate the participants’ mental health problems [28]. This further highlights the need to emphasize active coping behavior against DV.

Liang et al. (2005) proposed the following steps for seeking help in the case of DV: define the problem, decide whether to request help, and select a source of support. In this context, depending on the cultural values and norms, the range of coping strategies acceptable for abused women may differ [25]. For example, in a family-oriented culture, leaving the relationship may not be feasible for some women [25]. For others, confronting their partners or seeking outside assistance may mean going against their cultural norms and values [29]. Gender roles have recently changed due to rapid modernization and westernization in South Korea, altering the perception of DV [4]. Although women’s educational qualifications and economic activities have increased, their traditional values are weakening. On the other hand, because many men still exhibit and adhere to a patriarchal attitude, the role they play at home has not undergone much change, presenting a double burden for women [33]. This illustrates that unequal gender roles and patriarchal norms remain prevalent among families [34]. This could be the reason why most female victims showed passive coping behaviors in our study. Some Asian studies reported that women sought help only when the level of violence became critical or severe [24,25]. Although the prominence of the dominant ideology has faded over time, Confucian values and patriarchal familism are still prevalent in South Korea [23]. Consequently, the emphasis on family authority and honor over an individual’s happiness and safety negatively affects coping behavior strategies, hindering an abused woman’s decision to seek help [23,32]. Since these findings indicate that abused South Korean women generally use passive coping behaviors for a long period until they reach a dangerous point, proper early intervention is necessary.

Conversely, approximately only one-fifth of abused women reported or requested help during or after the time of violence. A relatively high frequency of informal help was requested from family members and friends compared to the frequency of formal help requested from the police and DV counseling centers. This finding was consistent with numerous previous studies that reported that a high rate of abused women sought personal help first [9,13,32,35]. There are inconsistent opinions regarding the effectiveness of personal help. Even at the scene of violence, neighbors and relatives were not helpful owing to their ignorance regarding DV [31,32]. Furthermore, the victims were led to formal help only if someone from among the family members or friends had experience or knowledge of DV [36]. Liang et al. stated that the support of friends or family members could, in fact, be an obstacle for the victim to request formal help [25]. Therefore, it is necessary to promote the capacity of informal social support systems used by abused women and provide them with opportunities to use formal support systems that may provide appropriate responses.

According to the stage model, IPV survivors initially coped with abuse privately, then gradually exhibited informal support-seeking behavior, and, as the violence worsened, sought more public/formal help [9,25]. Although abused women attempt various coping strategies depending on the situation, it is necessary for them to seek help from the formal system, which can help improve their coping strategies and reduce psychological distress [9,13]. Women who received formal support and services were twice as likely to be free of IPV in the subsequent two years compared to those who did not receive these services [25].

In the ecological model, the relationship level factors, such as type of DV, physical injuries, and feelings of intimidation or fear due to DV, were associated with the coping behavior of abused women. However, individual, community, and societal level factors were not associated with their coping behavior. In a study on IPV among Asian and Caucasian women, passive coping behavior was found to be a risk factor that increased their vulnerability to traumatic responses. However, perceived social support was a mediating factor that helped decrease psychological distress in Caucasians; it was not found to play a mediating role among Asians, which implied that the latter were more vulnerable to adverse health outcomes [8]. Such findings explain the lack of a significant impact of social support, such as community- and societal-level factors, on coping behaviors observed in this study. For example, the awareness of IPV-related laws and policies, such as “Non-family members (medical staff, neighbors, so on) can make a report to police if DV is suspected” and “The police must respond immediately on-scene to investigate”, was not at a high level in our study. This can be indicative of the fact that community or societal factors, such as support from neighbors or the police, DV-related laws, and policies, as well as support facilities and systems, have not been of practical help for women who are victims of DV. Based on these findings, comprehensive programs and policies that elicit effective coping behavior in abused women are required. In addition, social support needs to be established at the individual, relationship, community, and societal levels.

As primary prevention, age-appropriate educational programs from an early age on healthy relationships, gender equality, coping behaviors against violence, and organizations and methods to seek help are necessary [37]. In particular, in South Korea, there is a need to spread awareness of the nature and problems related to DV and promote family cultures that uphold mutual respect between the genders. Regarding secondary prevention, since health professionals are the first to come in contact with female victims, they should, initially, be able to identify abused women, provide the necessary care, and refer them to formal support services [37]. In particular, the risk perception of abused women should be recognized as an important sign of increased risk, and strong advocacy and risk reduction strategies should be applied, along with an assessment of fear [38]. Effective interventions in the early stages of abuse can reduce the negative effects of IPV and, consequently, reduce the likelihood of fatal violence against women [24]. Therefore, health professionals need to be educated about and trained on DV to strengthen their capacity to respond to female DV victims. Our findings will ultimately help in the development of comprehensive programs to respond to and prevent IPV.

## 5. Limitations and Prospect

This study has some limitations. First, since this study was an SDA, the items on coping behavior and factors at each level were limited, and the causal relationship could not be deduced. Second, the use of national data based on the Act on the Prevention of Domestic Violence and Protection, among others, limited the sample to only those in legal marital or de facto relationships. Thus, this study did not include adults who were divorced or dating. Third, we could not discuss the “asked others for help” aspect owing to the exclusion of the data for low frequency. Thus, further studies are required to examine those who used active coping strategies against DV. Lastly, the 2016 DVS includes a DV-related survey item entailing the act of violence being performed in a face-to-face manner, which was the focus of this study; our study did not consider DV occurring through non-face-to-face channels such as social networks and cellphone communication. It may be necessary to investigate such DV in the future.

## 6. Conclusions

Women who were victims of DV often “did nothing” rather than “escape the scene of violence or run outside” or “become reciprocally violent” in a DV situation. Compared to the “did nothing” group, the “escaped the scene of violence or ran outside” group was more likely to have experienced sexual abuse, physical injury, and felt intimidated or fearful due to DV. Since the family-oriented, patriarchal South Korean culture makes it difficult for DV victims to get practical support at the community or societal level, victims often choose passive coping behaviors until they suffer serious physical or psychological harm. Early intervention is important, as violence can gradually increase in severity if neglected. Since formal support helps with the improvement of coping strategies and problem-solving, there is a need to identify abused women through an active initial assessment, followed by a referral to formal support systems when required. In addition, this study indicates that only relationship-level factors influence coping behavior in the ecological model of DV. Therefore, policies and interventions need to be strengthened at multiple levels to help abused women appropriately cope with DV. Specifically, health professionals need to strengthen their ability to identify female DV victims early and to respond to them so that they can help them adopt appropriate coping behaviors. At the community and societal level, various education and publicity programs should be rolled out to cultivate a healthy attitude and awareness of gender roles and DV. In addition to related laws and policies, support resources should be actively promoted.

## Figures and Tables

**Figure 1 healthcare-10-00622-f001:**
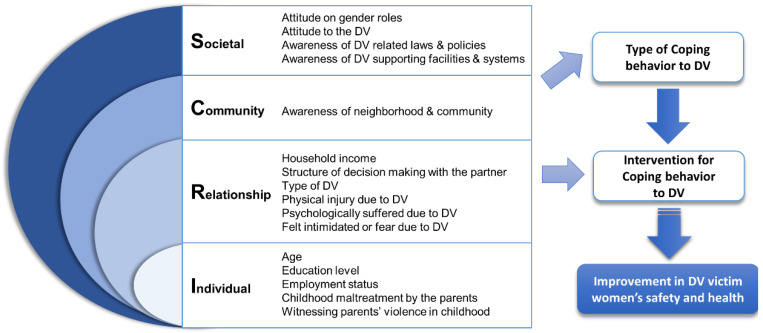
Conceptual framework. DV-Domestic Violence.

**Figure 2 healthcare-10-00622-f002:**
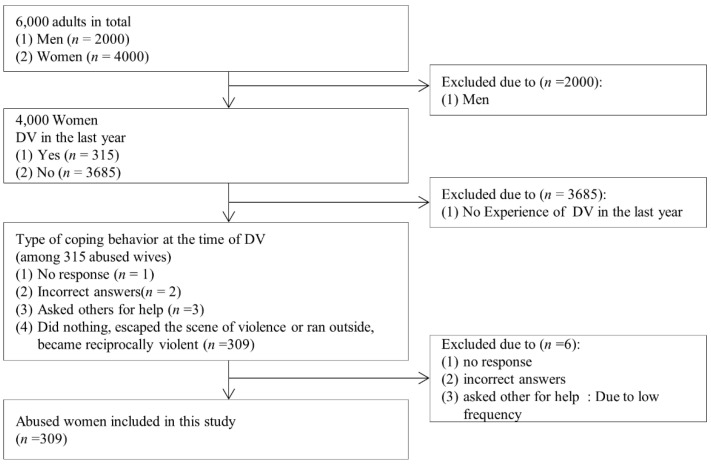
Flow diagram of the study sample. DV-Domestic Violence.

**Figure 3 healthcare-10-00622-f003:**
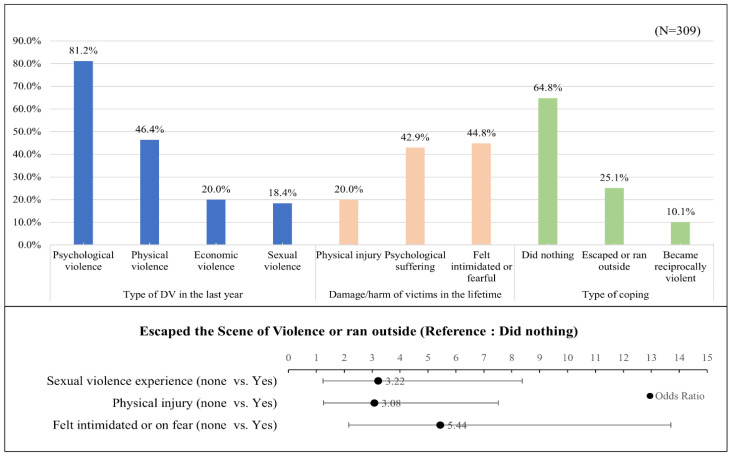
The main results of this study.

**Table 1 healthcare-10-00622-t001:** Group differences in multi-level factors by three coping behaviors (*n* = 309).

Ecolog-ical Model	Variable	Categories		Weighted % or Mean (SD)				χ^2^/F
				Total	Did Nothing *n* = 198 (64.8%)	Escaped the SOV or Ran Outside *n* = 77 (25.1%)	Became Reciproc-Ally Violent *n* = 34 (10.1%)	
I	Age (years)	19–39		20.4	63.3	16.3	20.4	9.64 *
		40–59		57.5	65.2	25.4	9.4	
		60 or more		22.1	64.1	32.1	3.8	
	Education level	At least middle school		19.7	70.2	27.7	2.1	9.13
		High school		56.1	64.2	26.9	8.9	
		College or higher		24.2	62.0	19.0	19.0	
	Employment status	Employed		51.5	60.2	26.8	13.0	3.38
		Unemployed		48.5	69.8	23.3	6.9	
	Childhood maltreatment by parents		Yes	78.7	64.4	25.0	10.6	0.35
			No	21.3	66.7	25.5	7.8	
	Witnessing violence among parents in childhood		Yes	61.7	60.1	28.4	11.5	3.37
			No	38.3	71.7	19.6	8.7	
R	Household income	Less than 4,000,000 won		52.7	64.3	28.6	7.1	3.52
		4,000,000 won or more		47.3	65.5	21.2	13.3	
	Structure of decision making with the partner			0.20 (0.53)	0.22 (0.53)	0.09 (0.47)	0.31 (0.57)	1.84
	Type of DV in the last year	Psychological violence	Yes	81.2	61.9	26.3	11.8	5.38
			No	18.8	77.8	20.0	2.2	
		Physical violence	Yes	46.4	51.4	35.1	13.5	16.62 **
			No	53.6	76.6	16.4	7.0	
		Economic violence	Yes	20.0	70.8	20.8	8.4	1.03
			No	80.0	63.0	26.1	10.9	
		Sexual violence	Yes	18.4	47.7	40.9	11.4	7.90 *
			No	81.6	68.7	21.5	9.8	
	Physical injury due to DV		Yes	20.0	29.2	56.2	14.6	36.09 **
			No	80.0	73.4	17.2	9.4	
	Psychological suffering due to DV		Yes	42.9	47.0	41.2	11.8	28.78 **
			No	57.1	78.7	12.5	8.8	
	Felt intimidated or fearful due to DV		Yes	44.8	44.9	43.9	11.2	39.54 **
			No	55.2	81.1	9.8	9.1	
C	Awareness of neighborhood and community			2.60 (0.49)	2.61 (0.49)	2.63 (0.48)	2.49 (0.56)	0.75
S	Attitude on gender roles			2.25 (0.55)	2.24 (0.54)	2.36 (0.54)	2.05 (0.58)	2.99
	Attitude toward DV			1.93 (0.39)	1.94 (0.38)	1.93 (0.40)	1.84 (0.39)	0.81
	Awareness of DV-related laws and policies			0.61 (0.49)	0.63 (0.32)	0.62 (0.27)	0.73 (0.27)	1.27
	Awareness of DV-supporting facilities			0.59 (0.30)	0.58 (0.31)	0.60 (0.29)	0.66 (0.29)	0.81

Notes: S: societal; C: community; R: relationship; I: individual; SD, standard deviation; SOV, scene of violence; won = Korean currency; ** *p* < 0.001, * *p* < 0.05.

**Table 2 healthcare-10-00622-t002:** Reasons for “did nothing” at the time of DV and past experience of asking for external help (*n* = 309).

Variable	Categories	Weighted %
Reasons for “did nothing” at the time of DV (Priority order)	Only needed to endure the DV for the moment	28.6
	He/she is my spouse	21.9
	Feeling ashamed	16.1
	If I respond, the violence gets worse	15.1
	Thinking of my child	6.8
	Thinking that it is my fault	5.2
	Feeling afraid	5.1
	Others	1.3
Past experience of asking for external help	No	79.6
	Yes	20.4
External helper ^†^ (multiple choice allowed)	Family or relatives	75.6
	Neighbors or friends	64.0
	Police	11.7
	Religious leaders	8.1
	Shelter for DV victims	4.8
	The 1366 hotline	4.8

Notes: DV, domestic violence. ^†^ Reported by subjects who sought external help.

**Table 3 healthcare-10-00622-t003:** Factors associated with DV-related coping behaviors (*n* = 309).

IV	Escaped the Scene of Violence or Ran Outside		Became Reciprocally Violent	
	OR [95% CI]	*p*	OR [95% CI]	*p*
Age (ref. = 19–39 years)				
40–59	1.52 [0.54–4.30]	0.434	0.69 [0.24–1.98]	0.489
60 or more	2.05 [0.51–8.25]	0.311	0.68 [0.09–4.90]	0.701
Education level (ref. = up to middle school graduates)				
High school graduates	1.44 [0.50–4.13]	0.498	4.67 [0.38–57.00]	0.227
College graduates or higher	1.54 [0.39–6.04]	0.537	10.47 [0.74–147.76]	0.082
Physical violence experience (ref. = none)				
Yes	1.45 [0.68–3.11]	0.336	2.32 [0.86–6.23]	0.095
Sexual violence experience (ref. = none)				
Yes	3.22 [1.24–8.37]	0.016	1.62 [0.48–5.50]	0.437
Physical injury (ref. = none)				
Yes	3.08 [1.26–7.52]	0.014	2.34 [0.68–8.11]	0.179
Psychological suffering (ref. = none)				
Yes	1.40 [0.58–3.39]	0.458	0.97 [0.31–3.03]	0.955
Felt intimidated or on fear (ref. = none)				
Yes	5.44 [2.16–13.70]	<0.001	2.11 [0.67–6.67]	0.202
Attitude on gender roles	1.41 [0.71–2.80]	0.323	0.76 [0.31–1.84]	0.543
Awareness of DV related laws and policies	0.60 [0.14–2.50]	0.480	2.38 [0.36–15.54]	0.366
Awareness of DV supporting facilities	2.56 [0.56–11.60]	0.224	1.34 [0.22–8.25]	0.751
Cox and snell = 0.299, Negelkerke = 0.364, McFadden = 0.206 (*p* < 0.001)				

Notes: IVs, independent variables; OR, odds ratio; 95% CI, 95% confidence interval; Ref., reference group. Multinomial reference group = “Did nothing” group.

## Data Availability

Data were obtained from the Ministry of Gender Equality in Korea and Family and are available from the Ministry of Gender Equality and Family with permission.

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
