# Peer review of "Factors Associated with Coping Behaviors of Abused Women: Findings from the 2016 Domestic Violence Survey"

_healthcare, 2022, doi:10.3390/healthcare10040622_

Round 1
Reviewer 1 Report
The manuscript healthcare-1624072 is devoted to the study of Intimate Partner Violence (IPV) as important public health problem. The reviewed article is interesting for scholars and theme of the article meets the scope of the journal. Work is performed at sufficient scientific level; the results of study are professionally interpreted. The manuscript may be considered for publication after minor revision in Healthcare. Besides quality and perception improving of the manuscript I would suggest to pay attention to the following notes:
- The style of some references in the Introduction should be changed. In some cases, there are from 4 to 7 sources after one sentence (lines 58, 59). This is unacceptable for publications in high-rated journals. Instead such references, it would be better to make a cross-reference discussion.
- It would be good to broaden the conclusions in the context of a more detailed presentation of ways to resolve the problem. I propose to divide Discussion section and add separate section "Limitations and prospects …."
- There are some numerous grammar and orthographical errors in the manuscript.
My decision is minor revision
Author Response
Response to Reviewer 1 Comments
Point 1: The style of some references in the Introduction should be changed. In some cases, there are from 4 to 7 sources after one sentence (lines 58, 59). This is unacceptable for publications in high-rated journals. Instead such references, it would be better to make a cross-reference discussion.
Response 1: The references have been represented by subject in the form of a literature review (page 2, line#62~67)
Point 2: It would be good to broaden the conclusions in the context of a more detailed presentation of ways to resolve the problem. I propose to divide Discussion section and add separate section "Limitations and prospects ….".
Response 2: I have divided the Discussion section into the discussion subsection and the limitations and prospects subsection.
Point 3: There are some numerous grammar and orthographical errors in the manuscript.
Response 3: The paper has gone through professional editing at a reputable organization and we have been presented a certificate of confirmation.

Reviewer 2 Report
I would like to thank the editors of this journal for the opportunity to give my point of view on this interesting research project. On the other hand, I would like to congratulate the authors of this research for the excellent field work they have done and the rigorous interpretation and presentation of the data and results.
I consider that the study is well laid out and the statistics are very rigorous, I would like to contribute some comments that I believe can be addressed by the authors and improve some aspects of the study.
It is not attached or indicated in the study the approval of the ethics committee and does not indicate whether it follows the rules of the declaration of Helsinki.
It does not understand that the study understands that the act of violence is performed in a face-to-face manner, but the study does not consider other non-face-to-face channels such as social networks. This can be contemplated as a limitation of the study, as it can be a good justification for future studies and to be taken into account in planning educational proposals to address the problem. Here are some references to address this in the discussion.
DOI: 10.5354/0719-1529.2015.32375
http://hdl.handle.net/11162/160074
https://doi.org/10.51698/aloma.2021.39.1.27-35
Regarding limitations of the study, cultural conditioning factors can also be contemplated, which have to be taken into account when taking this study as a reference for other populations.
Author Response
Response to Reviewer 2 Comments
Point 1: It is not attached or indicated in the study the approval of the ethics committee and does not indicate whether it follows the rules of the declaration of Helsinki.
Response 1: We have now outlined our research approval and ethics-related details in subsection “2.3 Ethical Considerations” and revised the relevant text with reference to the Author guidelines (page5, line # 202~203).
Point 2 : It does not understand that the study understands that the act of violence is performed in a face-to-face manner, but the study does not consider other non-face-to-face channels such as social networks. This can be contemplated as a limitation of the study, as it can be a good justification for future studies and to be taken into account in planning educational proposals to address the problem. Here are some references to address this in the discussion.
DOI: 10.5354/0719-1529.2015.32375
http://hdl.handle.net/11162/160074
https://doi.org/10.51698/aloma.2021.39.1.27-35
Regarding limitations of the study, cultural conditioning factors can also be contemplated, which have to be taken into account when taking this study as a reference for other populations.
Response 2: In accordance with your comment, we have added the following sentences in the Limitations and Prospects subsection.
Lastly, the 2016 DVS includes a DV-related survey item entailing the act of violence being performed in a face-to-face manner, which was the focus of this study. That is, our study did not consider DV occurring through non-face-to-face channels such as social networks and cellphone communication. It may be necessary to investigate such DV In the future. (page 12-13, line # 464~468).

Reviewer 3 Report
- On page 4 (line# 119-120), the authors will provide the internal consistency score of the CTS2.
- On page 4 (line# 128-129), the authors will provide the internal consistency score of the 8-item scale.
- On page 4 (line# 113-137), the authors will provide internal consistency scores for 7-item scale for “attitude toward gender roles”, 10-item scale for “attitude toward DV”, and 6-item scale for “awareness of related laws and policies and supporting facilities”.
- With internal consistency scores of the scales above, it will be difficult to judge the reliabilities of these scales.
- On page 8 (line# 214-218), the authors’ interpretations of obtained ORs were wrong. Since the ORs for reference category are 1, the correct interpretation should be “…2.22 times more likely to run away…” and “…fear due to DV were 2.08…and 4.44 times…more likely to run away…”.
- At the bottom of page 8 and top of page 9, in addition to the statement “…proportion of emotional abuse was highest across all four time points, follow by physical abuse”, the authors will mention that woman victims were likely suffered more than one type of abuse.
- On page 9 (line# 270-272), regarding the statement “…it is necessary to secure the safety of women upon their return home after a short leave, owing to financial and social poverty”. Do these women have other helps after they escape from the abusive partners? Family, friends, or shelter? They may not seek such external resources because of shame. Moreover, does the Korean culture discourage family and friends from helping or interfering? Does the Korean society promote enough public education on DV and women shelters?
- On page 10 (line# 301-302), statement “…escaped the scene of violence or ran outside...” can be a survival instinct for staying away from danger and being safe; it is not necessary a "passive" coping behavior.
- On page 11 (line# 357-359), ”…lack of a significant impact of social support, such as community-and societal-level factors, on coping behaviors observed in this study” need some clarification, are physicians required report the abuse to the police in Korea?
- On page 11 (line# 370-372), since “health professionals are the first to come in contact with female victims, they should, initially, be able to identify abused women, provide the necessary care, and refer them to formal support services”, health professionals may need the education about DV.
- In the Conclusions, authors will need to add specific examples for societal, policy, and professional practice changes.
Author Response
Response to Reviewer 3 Comments
Point 1: On page 4 (line# 119-120), the authors will provide the internal consistency score of the CTS2..
Response 1: We have presented Cronbach's alpha values on page 5 (line# 147-148) to provide the internal consistency score.
Point 2: On page 4 (line# 128-129), the authors will provide the internal consistency score of the 8-item scale.
Response 2: We have presented Cronbach's alpha values on page 5 (line# 166-167) to provide the internal consistency score.
Point 3: On page 4 (line# 113-137), the authors will provide internal consistency scores for 7-item scale for “attitude toward gender roles”, 10-item scale for “attitude toward DV”, and 6-item scale for “awareness of related laws and policies and supporting facilities”.
Response 3: We have presented Cronbach's alpha values on page 5 (line# 175-176, 181-182, 189-190,192-193) to provide the internal consistency score.
Point 4: With internal consistency scores of the scales above, it will be difficult to judge the reliabilities of these scales.
Response 4: This study was conducted based on national statistics from a survey, the 2016 DVS, conducted by a government agency. The said survey is an official statistical survey that is undergoing comprehensive quality evaluation to increase its reliability. We added the relevant content explaining this in the text (on page 5, line# 195~198).
Point 5: On page 8 (line# 214-218), the authors’ interpretations of obtained ORs were wrong. Since the ORs for reference category are 1, the correct interpretation should be “…2.22 times more likely to run away…” and “…fear due to DV were 2.08…and 4.44 times…more likely to run away…”.
Response 5: I appreciate your critique; however, I don't think our description of interpretations on this part is wrong.
If OR of the reference is 1 and the ORs of another variable is 1, it means that the probability between the reference and another variable is the same.
The interpretation was confirmed by referring to the references below, and another article in the "healthcare" journal that presents ORs the same as ours.
- Field, A. (2009) Discovering Statistics Using SPSS. 3rd Edition, Sage Publications Ltd., London.
- https://doi.org/10.3390/healthcare10020314
Point 6: At the bottom of page 8 and top of page 9, in addition to the statement “…proportion of emotional abuse was highest across all four time points, follow by physical abuse”, the authors will mention that woman victims were likely suffered more than one type of abuse.
Response 6: In accordance with your comment, the following sentence was added. (page 9, Line#296~297)
“Due to female DV victims being likely to suffer more than one type of abuse, duplicate responses were possible.”
Point 7: On page 9 (line# 270-272), regarding the statement “…it is necessary to secure the safety of women upon their return home after a short leave, owing to financial and social poverty”. Do these women have other helps after they escape from the abusive partners? Family, friends, or shelter? They may not seek such external resources because of shame. Moreover, does the Korean culture discourage family and friends from helping or interfering? Does the Korean society promote enough public education on DV and women shelters?
Response 7: In accordance with your comment, the following sentence was added. (page 10, Line#340~343)
Shelters for DV victims provide accommodation, meals, and counseling-oriented services [22], and private supporters such as friends and family provide emotional support depending on their level of DV awareness [28]. There are few financial aids available, and countermeasures are also required [22].
Point 8: On page 10 (line# 301-302), the statement “…escaped the scene of violence or ran outside...” can be a survival instinct for staying away from danger and being safe; it is not necessary a "passive" coping behavior.
Response 8: The sentence was changed from "passive coping behavior” to the following sentence. (page 11, Line#372~373)
This was a temporary strategy for female victims’ safety, not an active coping behavior to solve the problem.
Point 9: On page 11 (line# 357-359), ”…lack of a significant impact of social support, such as community-and societal-level factors, on coping behaviors observed in this study” need some clarification, are physicians required report the abuse to the police in Korea?
Response 9: In response with your comment, the following sentence was added. (page 12, Line#431~434)
For example, the awareness of IPV-related laws and policies such as “Non-family members (medical staff, neighbors, so on) can report to police if DV is suspected,” and “Police must respond immediately on-scene to investigate,” were not at a high level in our study.
Point 10: On page 11 (line# 370-372), since “health professionals are the first to come in contact with female victims, they should, initially, be able to identify abused women, provide the necessary care, and refer them to formal support services”, health professionals may need the education about DV.
Response 10: As per your comment, the following sentence was added. (page 12, Line# 451~453)
Therefore, health professionals need to be educated about and trained on DV to strengthen their capacity to respond to female DV victims.
Point 11: In the Conclusions, authors will need to add specific examples for societal, policy, and professional practice changes.
Response 11: In accordance with your comment, the following sentence was added. (page 13, Line# 482~488)
Therefore, policies and interventions need to be strengthened at multiple levels to help abused women appropriately cope with DV. Specifically, health professionals need to strengthen their ability to identify female DV victims early and to respond to them so that they can help them adopt appropriate coping behaviors. At the community and societal level, various education and publicity programs should be rolled out to cultivate a healthy attitude and awareness of gender roles and DV. In addition to related laws and policies, support resources should be actively promote.

Reviewer 4 Report
I would like to thank you for the opportunity to review this manuscript. The paper aims to identify the factors associated with the coping behaviors of abused women in South Korea, through a cross-sectional study among women who experienced domestic violence.
I am glad the authors brought this to the paper, which I found clear and well written.
Abstract: I found the abstract of the paper well-structured and clear, giving the reader all the relevant information. I’d like to suggest the authors add subheadings (background, methods, results) to make it clearer.
Introduction: The introduction is well written and concise, giving the reader all the essential background information for understanding the paper. However, I was asking myself if North and South Korea have the same patriarchal culture (e.g., lines 43-44) or if any differences are present. Since data refer to South Korea, a more specific literature overview would be helpful.
Method:
#1 I suggest removing paragraph 2.1 and adding the information about the design to paragraph 2.2
#2 I suggest using subheadings for scales and measures, to better understand them and adding an example item for each measure.
Results: Results are clear and well-written, I found them easy to follow.
Discussion: The discussion section clearly summarizes the findings and well-links them to the introduction, with a good job of elaborating and citing references; moreover, limitations are well addressed.
I’d like to suggest a minor revision to improve the paper and I congratulate the authors on addressing a very important problem. I hope that my comments are constructive and helpful.
Author Response
Response to Reviewer 4 Comments
Point 1: Abstract: I found the abstract of the paper well-structured and clear, giving the reader all the relevant information. I’d like to suggest the authors add subheadings (background, methods, results) to make it clearer.
Response 1: I have added subheadings (background, methods, results, conclusions) in the Abstract.
Point 2: Introduction: The introduction is well written and concise, giving the reader all the essential background information for understanding the paper. However, I was asking myself if North and South Korea have the same patriarchal culture (e.g., lines 43-44) or if any differences are present. Since data refer to South Korea, a more specific literature overview would be helpful.
Response 2: Page 5(line #45~50), I have revised mentions of “Korea” to “South Korea.”
Historically, North and South Korea had the same patriarchal culture. However, since the Korean War in 1950, North Korea has been a closed society while South Korea has been a liberal democracy. Therefore, it is assumed that there are many differences in terms of culture, including cultural issues related to IPV. Nonetheless, since this was not the aim of this study, additional literature on the matter was not included.
Point 3: Method:
#1 I suggest removing paragraph 2.1 and adding the information about the design to paragraph 2.2
Response 3: I have removed paragraph 2.1 and added the information about the design to paragraph 2.2 (page 3-5, line #100-193)
Point 4: #2 I suggest using subheadings for scales and measures, to better understand them and adding an example item for each measure.
Response 4: I have added subheadings for scales and measures following your recommendations, and examples for each measurement item were added.

Reviewer 5 Report
The results of an extremely important problem of family violence are presented in the manuscript. The authors, based on specific data (large cross-sections) for several years, analyzed the prevalence of violence (which acquires catastrophic proportions within one particular country), the "weight" of each specific type of violence. Note that in the modern world, not only women become victims, but men are also often subject to "abuse" by women, especially in the case of adherence to the Western cultural model. Therefore, it is necessary to mention this in the article. The authors noted that there is a cultural "norm" for a certain type of coping of victims of violence (humility). However, this is characteristic not only of Confucian culture, but also of other cultures. This makes the article interesting for representatives of other cultures as well. Nevertheless, there are several recommendations for improving the perception of the article. It is unclear from the article what factors the authors identified using using data from the 2016 National Survey? It is necessary to indicate that we are talking about specific demographic data already in the goals and objectives (otherwise the reader of the entire text is waiting for the appearance of psychological factors). What hypotheses were tested during the research? You must also specify. In addition, it is necessary to clearly present the conclusions: which as a result is new theoretical knowledge .
Author Response
Response to Reviewer 5 Comments
Point 1: Note that in the modern world, not only women become victims, but men are also often subject to "abuse" by women, especially in the case of adherence to the Western cultural model. Therefore, it is necessary to mention this in the article.
Response 1: On page 1 (line #31-32), I have added the following sentence.
Although both men and women can be IPV victims and perpetrators, IPV against women refers to …
Point 2: It is unclear from the article what factors the authors identified using data from the 2016 National Survey? It is necessary to indicate that we are talking about specific demographic data already in the goals and objectives (otherwise the reader of the entire text is waiting for the appearance of psychological factors).
Response 2: The following contents were added to the front of the description of the aim (page 2, line#88-90).
The subjects in this study were females who had recently experienced DV as per the 2016 DVS. Among the variables investigated in the 2016 DVS, variables included in each level of the ecological model were selected.
Point 3: What hypotheses were tested during the research? You must also specify.
Response 3: On page 2 (line #73-74), I have added the following sentence.
Our research question was what coping strategies do female DV victims in South Korea use and what factors influence those coping strategies?
Point 4: it is necessary to clearly present the conclusions: which as a result is new theoretical knowledge.
Response 4: In answering our research question, I have included our conclusions about the main coping strategies used by female IPV victims in South Korea and factors that influence these coping strategies in the Conclusion section. Also, I have added the following sentence. (page 13, Line#482~488)
Therefore, policies and interventions need to be strengthened at multiple levels to help abused women appropriately cope with DV. Specifically, health professionals need to strengthen their ability to identify female DV victims early and to respond to them so that they can help them adopt appropriate coping behaviors. At the community and societal level, various education and publicity programs should be rolled out to cultivate a healthy attitude and awareness of gender roles and DV. In addition to related laws and policies, support resources should be actively promoted.
